# Deep Positive-Unlabeled Anomaly Detection for Contaminated Unlabeled Data

## Abstract

Semi-supervised anomaly detection, which aims to improve the anomaly detection performance by using a small amount of labeled anomaly data in addition to unlabeled data, has attracted attention. Existing semi-supervised approaches assume that most unlabeled data are normal, and train anomaly detectors by minimizing the anomaly scores for the unlabeled data while maximizing those for the labeled anomaly data. However, in practice, the unlabeled data are often contaminated with anomalies. This weakens the effect of maximizing the anomaly scores for anomalies, and prevents us from improving the detection performance. To solve this problem, we propose the deep positive-unlabeled anomaly detection framework, which integrates positive-unlabeled learning with deep anomaly detection models such as autoencoders and deep support vector data descriptions. Our approach enables the approximation of anomaly scores for normal data using the unlabeled data and the labeled anomaly data. Therefore, without labeled normal data, our approach can train anomaly detectors by minimizing the anomaly scores for normal data while maximizing those for the labeled anomaly data. Experiments on various datasets show that our approach achieves better detection performance than existing approaches.

## 1. Introduction

Anomaly detection, which aims to identify unusual data points, is an important task in machine learning (Ruff et al., 2021). It has been performed in various fields such as cyber-security (Kwon et al., 2019), novelty detection (Marchi et al., 2015), medical diagnosis (Litjens et al., 2017), infrastructure monitoring (Borghesi et al., 2019), and natural sciences (Min et al., 2017; Cerri et al., 2019; Pracht et al., 2020).

In general, anomaly detection is performed by unsupervised learning because it does not require expensive and time-consuming labeling. Unsupervised approaches assume that most unlabeled data are normal, and try to detect anomalies by using an anomaly score, which represents the difference from normal data (Hinton & Salakhutdinov, 2006; Ruff et al., 2018). Although these approaches are easy to handle, their detection performance is limited because they cannot use information about anomalies.

To improve the detection performance, semi-supervised anomaly detection uses a small amount of labeled anomaly data in addition to unlabeled data. Existing semi-supervised approaches train anomaly detectors to minimize the anomaly scores for the unlabeled data, and to maximize those for the labeled anomaly data (Hendrycks et al., 2018; Ruff et al., 2019; Yamanaka et al., 2019). However, in practice, the unlabeled data are often contaminated with anomalies. This weakens the effect of maximizing the anomaly scores for anomalies, and prevents us from improving the anomaly detection performance. This frequently occurs because it is difficult to label all anomalies.

Our purpose is to propose a semi-supervised approach that can improve the anomaly detection performance even if the unlabeled data are contaminated with anomalies. To handle such unlabeled data, we propose the deep positive-unlabeled anomaly detection framework, which integrates positive-unlabeled (PU) learning (Du Plessis et al., 2014; 2015; Kiryo et al., 2017) with deep anomaly detectors such as the autoencoder (AE) (Hinton & Salakhutdinov, 2006) and the deep support vector data description (DeepSVDD) (Ruff et al., 2018). PU learning assumes that an unlabeled data distribution is a mixture of normal and anomaly data distributions[1]. Accordingly, the normal data distribution is approximated by using the unlabeled and anomaly data distributions. With this assumption, we approximate the anomaly scores for normal data using the unlabeled data and the labeled anomaly data. Therefore, without labeled normal data, we can train anomaly detectors to minimize the anomaly scores for normal data, and to maximize those for the labeled anomaly data.

---

[1]Anonymous Institution, Anonymous City, Anonymous Region, Anonymous Country. Correspondence to: Anonymous Author <anon.email@domain.com>.

Preliminary work. Under review by the International Conference on Machine Learning (ICML). Do not distribute.

---

[1]Note that the anomaly data in the training dataset follow the anomaly data distribution, but new types of anomalies, unseen during training, may NOT follow this distribution. In general, no distribution can fully represent all possible anomalies.

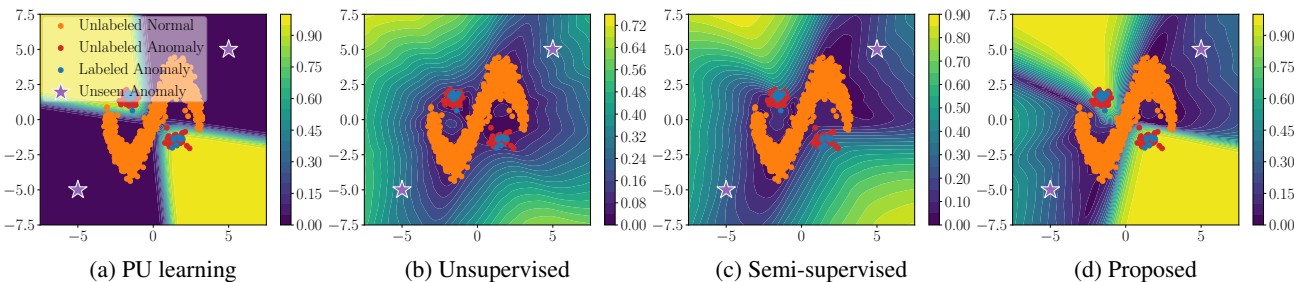

| (a) PU learning | (b) Unsupervised | (c) Semi-supervised | (d) Proposed |

Figure 1: The comparison of the PU learning, the unsupervised anomaly detector (DAE), the semi-supervised anomaly detector (ABC), and the proposed method on the toy dataset. The unlabeled data in this dataset include both normal and anomaly data points. The yellow and blue in the contour maps represent abnormality and normality, respectively. The purple stars represent the examples of unseen anomalies, which are new types of anomalies unseen during training.

Figure 1 compares the PU learning (Kiryo et al., 2017), the unsupervised detector, the semi-supervised detector, and our approach on the toy dataset. We used the denoising AE (DAE) (Vincent et al., 2008) for the unsupervised detector, and the autoencoding binary classifier (ABC) (Yamanaka et al., 2019) for the semi-supervised detector. Our approach is based on the DAE. The toy dataset consists of unlabeled and anomaly data, where the unlabeled data include both normal and anomaly data points. We first focus on the PU learning, which aims to train the binary classifier from the unlabeled data and the labeled anomaly data. This can detect *seen anomalies*, which are similar to anomalies included in the training dataset. However, since its decision boundary is between normal data points and seen anomalies, it cannot detect *unseen anomalies*, which are new types of anomalies unseen during training, such as novel anomalies and zero-day attacks (Wang et al., 2013; Pang et al., 2021; Ding et al., 2022). We next focus on unsupervised and semi-supervised detectors. They can detect unseen anomalies to some extent since they try to detect anomalies by using the difference from normal data. The unsupervised detector cannot detect seen anomalies since it cannot use information about anomalies. The semi-supervised detector can detect seen anomalies to some extent since it can use the labeled anomaly data. However, the contaminated dataset weakens the effect of maximizing the anomaly scores for anomalies in the semi-supervised detector. Finally, we focus on our approach. our approach can detect seen anomalies according to the effectiveness of PU learning, and can detect unseen anomalies to some extent according to the effectiveness of the deep anomaly detector.

Our framework is applicable to various anomaly detector. When selecting a detector, we require that its loss function be non-negative and differentiable. For example, the AE, the DeepSVDD, and recent self-supervised detectors such as (Hendrycks et al., 2019; Qiu et al., 2021; Shenkar & Wolf, 2021) satisfy this. In this paper, we apply our framework to the AE and the DeepSVDD. We refer to the former as

the positive-unlabeled autoencoder (PUAE), and the latter as the positive-unlabeled support vector data description (PUSVDD).

Compared to existing semi-supervised approaches designed to handle contaminated unlabeled data (Zhang et al., 2018; Ju et al., 2020; Zhang et al., 2021; Pang et al., 2023; Li et al., 2023; Perini et al., 2023), the proposed method is theoretically justified from the perspective of unbiased PU learning (Du Plessis et al., 2014; 2015; Kiryo et al., 2017). In addition, the proposed method achieved equal to or better performance than the current state-of-the-art approach (Li et al., 2023) across eight datasets.

Our contributions can be summarized as follows:

- To handle contaminated unlabeled data, we propose the deep positive-unlabeled anomaly detection framework, which integrates unbiased PU learning with the deep anomaly detectors such as the AE and the DeepSVDD.

- Experiments on various datasets demonstrate that our approach outperforms existing approaches in anomaly detection performance.

## 2. Preliminaries

In this section, we first explain our problem setup. Next, we review the AE (Hinton & Salakhutdinov, 2006) and the ABC (Yamanaka et al., 2019). They are typical unsupervised and semi-supervised anomaly detection approaches, and our framework can be applied to them.

### 2.1. Problem Setup

Given unlabeled dataset $\mathcal{U} = \{\mathbf{x}_1, \ldots, \mathbf{x}_N\}$ and anomaly dataset $\mathcal{A} = \{\tilde{\mathbf{x}}_1, \ldots, \tilde{\mathbf{x}}_M\}$ for training. $\mathcal{U}$ contains not only normal data points but also seen anomalies that are similar to $\mathcal{A}$. The test dataset contains normal data points, seen anomalies, and unseen anomalies that are new types of

anomalies unseen during training. Our goal is to obtain a high performance anomaly detector using $\mathcal{U}$ and $\mathcal{A}$.

## 2.2. Autoencoder

For unsupervised anomaly detectors, we train them only using the unlabeled dataset $\mathcal{U}$. As an example, we focus on the AE, which has been successfully applied to anomaly detection (Sakurada & Yairi, 2014). The AE is presented for representation learning, which learns the representation of data points through data reconstruction. Let $\mathbf{x}$ be a data point and $\mathbf{z}$ be its low-dimensional latent representation. The AE consists of two neural networks: encoder $E_\theta(\mathbf{x})$ and decoder $D_\theta(\mathbf{z})$, where $\theta$ is the parameter of these neural networks. $E_\theta(\mathbf{x})$ maps a data point $\mathbf{x}$ into a low-dimensional latent representation $\mathbf{z}$, and $D_\theta(\mathbf{z})$ reconstructs the original data point $\mathbf{x}$ from the latent representation $\mathbf{z}$. The reconstruction error for each data point $\mathbf{x}$ in the AE is defined as follows:

$$\ell(\mathbf{x};\theta) = \|D_\theta(E_\theta(\mathbf{x})) - \mathbf{x}\|, \quad (1)$$

where $\|\cdot\|$ represents $\ell_2$ norm.

When the AE is used for anomaly detection, all unlabeled data points are assumed to be normal. We train the AE by minimizing the following objective function:

$$\mathcal{L}_{\mathrm{AE}}(\theta) = \frac{1}{N} \sum_{n=1}^{N} \ell(\mathbf{x}_n;\theta). \quad (2)$$

After training, the AE is expected to successfully reconstruct normal data and fail to reconstruct anomaly data because the training dataset is assumed to contain only normal data and no anomaly data. Hence, the reconstruction error can be used for the anomaly score.

Although unsupervised approaches are widely used in anomaly detection, their detection performance is limited because they cannot use information about anomalies.

## 2.3. Autoencoding Binary Classifier

Semi-supervised anomaly detection aims to improve the anomaly detection performance using the unlabeled dataset $\mathcal{U}$ and the anomaly dataset $\mathcal{A}$. A number of studies have been presented such as the ABC (Yamanaka et al., 2019), the deep semi-supervised anomaly detection (DeepSAD) (Ruff et al., 2019), and the outlier exposure (Hendrycks et al., 2018). Here, we focus on the ABC, which is based on the AE.

Let $y = 0$ be normal and $y = 1$ be anomaly. The ABC models the conditional probability of $y$ given $\mathbf{x}$ by using the reconstruction error $\ell(\mathbf{x};\theta)$ as follows:

$$p_\theta(y|\mathbf{x}) = \begin{cases} \exp(-\ell(\mathbf{x};\theta)) & (y=0) \\ 1 - \exp(-\ell(\mathbf{x};\theta)) & (y=1) \end{cases}. \quad (3)$$

A small reconstruction error results in a higher probability of normality $p_\theta(y = 0|\mathbf{x})$, while a large reconstruction error results in a higher probability of abnormality $p_\theta(y = 1|\mathbf{x})$.

With this conditional probability, the ABC introduces the binary cross entropy as the loss function for each data point as follows:

$$\begin{aligned} \ell_{\mathrm{BCE}}(\mathbf{x}, y; \theta) &= -\log p_\theta(y|\mathbf{x}) \\ &= (1-y)\ell(\mathbf{x};\theta) - y\log(1 - \exp(-\ell(\mathbf{x};\theta))). \end{aligned} \quad (4)$$

Like the AE, the ABC assumes all unlabeled data points to be normal. The ABC is trained by minimizing the following objective function:

$$\begin{aligned} \mathcal{L}_{\mathrm{ABC}}(\theta) &= \frac{1}{N} \sum_{n=1}^{N} \ell_{\mathrm{BCE}}(\mathbf{x}_n, 0; \theta) \\ &\quad + \frac{1}{M} \sum_{m=1}^{M} \ell_{\mathrm{BCE}}(\tilde{\mathbf{x}}_m, 1; \theta). \end{aligned} \quad (5)$$

This minimizes the reconstruction errors for the unlabeled data and maximizes those for the anomaly data. Hence, after training, the AE becomes to reconstruct the unlabeled data assumed to be normal, and fail to reconstruct anomalies. Other semi-supervised anomaly detection approaches such as the DeepSAD (Ruff et al., 2019) and the outlier exposure (Hendrycks et al., 2018) also minimize the anomaly scores for the unlabeled data and maximize those for the anomaly data.

However, the unlabeled dataset $\mathcal{U}$ is often contaminated with anomalies in practice. This contamination weakens the effect of maximizing the anomaly scores for anomalies, and prevents us from improving the detection performance. This frequently occurs because it is difficult to label all anomalies in the unlabeled dataset.

## 3. Proposed Method

We aim to improve the detection performance even if the unlabeled dataset $\mathcal{U}$ contains anomalies. To handle such unlabeled dataset, we propose the deep positive-unlabeled anomaly detection framework, which integrates PU learning (Du Plessis et al., 2014; 2015; Kiryo et al., 2017) with deep anomaly detection models such as the AE, the DeepSVDD, and recent self-supervised detectors such as (Hendrycks et al., 2019; Qiu et al., 2021; Shenkar & Wolf, 2021).

In this section, we apply our framework to the AE and the DeepSVDD. We refer to the former as the positive-unlabeled autoencoder (PUAE), and the latter as the positive-unlabeled support vector data description (PUSVDD).

Hereinafter, we also refer anomalies as positive (+) samples, and normal data points as negative (-) samples.

### 3.1. Positive-Unlabeled Autoencoder

At first, we explain the PUAE. Let $p_\mathcal{N}$ be the normal data distribution, $p_\mathcal{A}$ be the seen anomaly distribution, and $p_\mathcal{U}$ be the unlabeled data distribution. We assume that the datasets $\mathcal{U}$ and $\mathcal{A}$ are drawn from $p_\mathcal{U}$ and $p_\mathcal{A}$, respectively. We also assume that $p_\mathcal{U}$ can be rewritten as follows:

$$p_\mathcal{U}(\mathbf{x}) = \alpha p_\mathcal{A}(\mathbf{x}) + (1 - \alpha) p_\mathcal{N}(\mathbf{x}), \quad (6)$$

where $\alpha \in [0, 1]$ is the probability of anomaly occurrence in the unlabeled data. Hence, $p_\mathcal{N}$ can be rewritten as follows:

$$(1 - \alpha) p_\mathcal{N}(\mathbf{x}) = p_\mathcal{U}(\mathbf{x}) - \alpha p_\mathcal{A}(\mathbf{x}). \quad (7)$$

Although $\alpha$ is the hyperparameter and is assumed to be known throughout this paper, it can be estimated from the datasets $\mathcal{U}$ and $\mathcal{A}$ in conventional PU learning approaches (Menon et al., 2015; Ramaswamy et al., 2016; Jain et al., 2016; Christoffel et al., 2016).

If we have access to the normal data distribution $p_\mathcal{N}$, we can train the AE by minimizing the following ideal objective function:

$$\begin{aligned} \mathcal{L}_{\mathrm{PN}}(\theta) = {} & \alpha \mathbb{E}_{p_\mathcal{A}}[\ell_{\mathrm{BCE}}(\mathbf{x}, 1; \theta)] \\ & + (1 - \alpha) \mathbb{E}_{p_\mathcal{N}}[\ell_{\mathrm{BCE}}(\mathbf{x}, 0; \theta)], \end{aligned} \quad (8)$$

where $\mathbb{E}[\cdot]$ is the expectation. Since we cannot access $p_\mathcal{N}$ in practice, we have to approximate the second term in Eq. (8). According to Eq. (7), this can be rewritten as follows:

$$\begin{aligned} & (1 - \alpha) \mathbb{E}_{p_\mathcal{N}}[\ell_{\mathrm{BCE}}(\mathbf{x}, 0; \theta)] \\ & = \mathbb{E}_{p_\mathcal{U}}[\ell_{\mathrm{BCE}}(\mathbf{x}, 0; \theta)] - \alpha \mathbb{E}_{p_\mathcal{A}}[\ell_{\mathrm{BCE}}(\mathbf{x}, 0; \theta)]. \end{aligned} \quad (9)$$

Hence, by using the seen anomaly distribution $p_\mathcal{A}$ and the unlabeled data distribution $p_\mathcal{U}$, $\mathcal{L}_{\mathrm{PN}}(\theta)$ can be rewritten as follows:

$$\begin{aligned} \mathcal{L}_{\mathrm{PN}}(\theta) = {} & \alpha \mathbb{E}_{p_\mathcal{A}}[\ell_{\mathrm{BCE}}(\mathbf{x}, 1; \theta)] \\ & + \mathbb{E}_{p_\mathcal{U}}[\ell_{\mathrm{BCE}}(\mathbf{x}, 0; \theta)] - \alpha \mathbb{E}_{p_\mathcal{A}}[\ell_{\mathrm{BCE}}(\mathbf{x}, 0; \theta)]. \end{aligned} \quad (10)$$

With the datasets $\mathcal{U}$ and $\mathcal{A}$, we can approximate $\mathcal{L}_{\mathrm{PN}}(\theta)$ by the empirical distribution as follows:

$$\mathcal{L}_{\mathrm{PN}}(\theta) \simeq \alpha \underbrace{\frac{1}{M} \sum_{m=1}^{M} \ell_{\mathrm{BCE}}(\tilde{\mathbf{x}}_m, 1; \theta)}_{\mathcal{L}_\mathcal{A}^+(\theta)}$$
$$+ \underbrace{\frac{1}{N} \sum_{n=1}^{N} \ell_{\mathrm{BCE}}(\mathbf{x}_n, 0; \theta)}_{\mathcal{L}_\mathcal{U}^-(\theta)} - \alpha \underbrace{\frac{1}{M} \sum_{m=1}^{M} \ell_{\mathrm{BCE}}(\tilde{\mathbf{x}}_m, 0; \theta)}_{\mathcal{L}_\mathcal{A}^-(\theta)}. \quad (11)$$

---

**Algorithm 1** Positive-Unlabeled Autoencoder

**Input**: Unlabeled and anomaly datasets $(\mathcal{U}, \mathcal{A})$, mini-batch size $K$, hyperparameter $\alpha \in [0, 1]$
**Output**: Model parameter $\theta$
**Procedure**:
1: **while** not converged **do**
2:     Sample mini-batch $\mathcal{B}$ from datasets $(\mathcal{U}, \mathcal{A})$
3:     Compute $\mathcal{L}_\mathcal{A}^+(\theta)$, $\mathcal{L}_\mathcal{U}^-(\theta)$, and $\mathcal{L}_\mathcal{A}^-(\theta)$ in Eq. (11) with $\mathcal{B}$
4:     Set the gradient $\nabla_\theta(\alpha \mathcal{L}_\mathcal{A}^+(\theta) + \left| \mathcal{L}_\mathcal{U}^-(\theta) - \alpha \mathcal{L}_\mathcal{A}^-(\theta) \right|)$
5:     Update $\theta$ with the gradient
6: **end while**

---

In this equation, the sum of the second and third terms is the approximation of the anomaly scores for normal data:

$$(1 - \alpha) \mathbb{E}_{p_\mathcal{N}}[\ell_{\mathrm{BCE}}(\mathbf{x}, 0; \theta)] \simeq \mathcal{L}_\mathcal{U}^-(\theta) - \alpha \mathcal{L}_\mathcal{A}^-(\theta). \quad (12)$$

The left-hand side in Eq. (12) is always greater than or equal to zero, but the right-hand side can be negative. In experiments, it often converges towards negative infinity, resulting in the meaningless solution. To avoid this, based on (Hammoudeh & Lowd, 2020), our training objective function to be minimized ensures that $\mathcal{L}_\mathcal{U}^-(\theta) - \alpha \mathcal{L}_\mathcal{A}^-(\theta)$ is not negative as follows:

$$\mathcal{L}_{\mathrm{Proposed}}(\theta) = \alpha \mathcal{L}_\mathcal{A}^+(\theta) + \left| \mathcal{L}_\mathcal{U}^-(\theta) - \alpha \mathcal{L}_\mathcal{A}^-(\theta) \right|. \quad (13)$$

We can optimize this training objective function by using the stochastic gradient descent (SGD) such as Adam (Kingma & Ba, 2015). We refer to this approach as the PUAE.

Algorithm 1 shows the pseudo code of the PUAE, where $K$ is the mini-batch size for the SGD. Note that our approach can be easily extended to positive-negative-unlabeled (PNU) learning (Sakai et al., 2017), where we can also use a small amount of labeled normal data.

### 3.2. Positive-Unlabeled Support Vector Data Description

We next apply our framework to the DeepSVDD. The DeepSVDD aims to pull the representation of the normal data towards the pre-defined center, and push those of the anomaly data away from the center. Let $f_\theta(\mathbf{x})$ be the feature extractor, like the encoder in the AE. The loss function for each data point of the DeepSVDD is defined as follows:

$$\tilde{\ell}(\mathbf{x}; \theta) = \| f_\theta(\mathbf{x}) - \mathbf{c} \|^2, \quad (14)$$

where $\mathbf{c} \neq \mathbf{0}$ is the pre-defined center vector.

The DeepSAD (Ruff et al., 2019) is a semi-supervised extension of the DeepSVDD. The DeepSAD trains the DeepSVDD model to minimize Eq. (14) for the unlabeled data, and to maximize it for the anomaly data. The loss

function for each data point of the DeepSAD is defined as follows:

$$\tilde{\ell}_{\mathrm{SAD}}(\mathbf{x}, y; \theta) = (1 - y)\tilde{\ell}(\mathbf{x}; \theta) + \frac{y}{\tilde{\ell}(\mathbf{x}; \theta)}. \quad (15)$$

We can apply our framework to the DeepSVDD by replacing Eq. (4) in the PUAE with Eq. (15), while keeping all other components identical to the PUAE. We refer to this approach as the PUSVDD.

### 3.3. Application to Other Anomaly Detectors

Our framework is applicable to anomaly detection models whose loss functions are non-negative and differentiable. As described above, the AE and the DeepSVDD satisfy this requirement. In addition, a lot of models satisfy this requirement, such as the DAE (Vincent et al., 2008) and recent self-supervised detectors (Hendrycks et al., 2019; Qiu et al., 2021; Shenkar & Wolf, 2021). For example, we focus on the DAE. The DAE is a variant of the AE, and has also been successfully applied to anomaly detection (Sakurada & Yairi, 2014). The DAE aims to reconstruct original data points from noisy input data points. Its loss function for each data point is defined as follows:

$$\ell(\mathbf{x}; \theta) = \|D_\theta(E_\theta(\mathbf{x} + \epsilon)) - \mathbf{x}\|, \quad (16)$$

where $\epsilon$ is a noise from an isotropic Gaussian distribution. Our framework can be applied to the DAE by substituting this into Eq. (1). In this way, our framework can be applied by substituting the loss function of the desired model into Eq. (1) in the PUAE or Eq. (14) in the PUSVDD.

## 4. Related Work

### 4.1. Unsupervised Anomaly Detection

Numerous unsupervised approaches have been presented, ranging from shallow approaches such as the one-class support vector machine (OCSVM) (Tax & Duin, 2004) and the isolation forest (IF) (Liu et al., 2008) to deep approaches such as the AE (Hinton & Salakhutdinov, 2006) and the DeepSVDD (Ruff et al., 2018). In addition, generative models such as the variational autoencoder (Kingma & Welling, 2014; Kingma et al., 2015) and the generative adversarial nets (Goodfellow et al., 2014) are also used for anomaly detection (Choi et al., 2018; Serrà et al., 2019; Ren et al., 2019; Perera et al., 2019; Xiao et al., 2020; Havtorn et al., 2021; Yoon et al., 2021). Although they are often used in anomaly detection, their detection performance is limited because they cannot use information about anomalies. For example, generative models may fail to detect anomalies that are obvious to the human eye (Nalisnick et al., 2018).

Furthermore, these approaches assume that unlabeled data are mostly normal. However, they are contaminated with anomalies in practice, degrading the detection performance. Several unsupervised approaches have been presented to handle such contaminated unlabeled data (Zhou & Paffenroth, 2017; Qiu et al., 2022; Shang et al., 2023). Among them, the latent outlier exposure (LOE) (Qiu et al., 2022) is a representative approach. The LOE introduces the label for each data point as the latent variable, and alternates between inferring the latent label and optimizing the parameter of the base anomaly detector. Compared to these approaches, our approach can achieve better detection performance by using the unlabeled data and the labeled anomaly data, even if the unlabeled data are contaminated with anomalies. As the base detector for our approach, the AE, the DeepSVDD, and recent self-supervised detectors (Hendrycks et al., 2019; Qiu et al., 2021; Shenkar & Wolf, 2021) can be used as described in Section 3. In addition, our approach can also be applied to the LOE by substituting its objective function into $\mathcal{L}_{\mathcal{U}}^{-}(\theta)$ in Eq. (11).

### 4.2. Semi-supervised Anomaly Detection

Several semi-supervised approaches have been presented, aiming to improve the detection performance using labeled anomaly data in addition to unlabeled data (Hendrycks et al., 2018; Yamanaka et al., 2019; Ruff et al., 2019). Compared to these approaches, our approach can effectively handle the unlabeled data that are contaminated with anomalies, as described in Section 3.1.

To handle contaminated unlabeled data, a number of semi-supervised approaches have been presented, including PU learning approaches (Zhang et al., 2018; Ju et al., 2020; Zhang et al., 2021; Pang et al., 2023; Li et al., 2023; Perini et al., 2023). Among them, the semi-supervised outlier exposure with a limited labeling budget (SOEL) (Li et al., 2023) is the current state-of-the-art approach. The SOEL is a semi-supervised extension of the LOE (Qiu et al., 2022), and presents the query strategy for the LOE, deciding which data should be labeled. Compared to these approaches, our approach is theoretically justified from the perspective of unbiased PU learning (Du Plessis et al., 2014; 2015; Kiryo et al., 2017). In addition, our approach achieved equal to or better performance than the SOEL across eight datasets, as described in Section 5.

### 4.3. Positive-Unlabeled Learning

A lot of PU learning approaches have been presented for binary classification (Elkan & Noto, 2008; Du Plessis et al., 2014; 2015; Kiryo et al., 2017; Bekker & Davis, 2020; Nakajima & Sugiyama, 2023). Among them, our approach is based on the unbiased PU learning (Du Plessis et al., 2014; 2015; Kiryo et al., 2017). The empirical risk estimators in Eq. (11) is unbiased and consistent with respect to all popular loss function. This means for fixed $\theta$, Eq. (11),

which is the approximation of Eq. (8), converges to Eq. (8) as the dataset sizes $N, M \rightarrow \infty$. It is known that if the model is linear with respect to $\theta$, a particular loss function will result in convex optimization, and the globally optimal solution can be obtained (Natarajan et al., 2013; Patrini et al., 2016; Niu et al., 2016). Despite this ideal property, when the model is complex such as neural networks, Eq. (11) can become negative, potentially leading to the meaningless solution. To address this issue, following (Hammoudeh & Lowd, 2020), we take its absolute value, as described in Section 3.1.

Compared to conventional PU learning that is based on the binary classifier, our approach is based on deep anomaly detection models such as the AE and the DeepSVDD. Although conventional PU learning cannot detect unseen anomalies since its decision boundary is between normal data points and seen anomalies, our approach can detect both seen and unseen anomalies.

## 5. Experiments

### 5.1. Data

We used following eight image datasets: MNIST (Salakhutdinov & Murray, 2008), FashionMNIST (Xiao et al., 2017), SVHN (Netzer et al., 2011), CIFAR10 (Krizhevsky et al., 2009), CIFAR100 (Krizhevsky et al., 2009), PathMNIST, OCTMNIST, and TissueMNIST (Yang et al., 2021; 2023).

First, we explain the first four datasets. MNIST is the handwritten digits, FashionMNIST is the fashion product images, SVHN is the house number digits, and CIFAR10 is the animal and vehicle images. We resized all datasets to $32 \times 32$ resolution. These datasets consist of 10 class images. For MNIST and SVHN, we used the digits as class indices. For FashionMNIST, we indexed labels as: {T-shirt/top: 0, Trouser: 1, Pullover: 2, Dress: 3, Coat: 4, Sandal: 5, Shirt: 6, Sneaker: 7, Bag: 8, and Ankle boot: 9}. For CIFAR10, we indexed labels as: {airplane: 0, automobile: 1, bird: 2, cat: 3, deer: 4, dog: 5, frog: 6, horse: 7, ship: 8, and truck: 9}. We extend the experiments in (Ruff et al., 2018) to semi-supervised anomaly detection with contaminated unlabeled data. Of the 10 classes, we used one class as normal, another class as unseen anomaly, and the remaining classes as seen anomaly. For example with MNIST, if we use the digit 1 as normal and the digit 0 as unseen anomaly, seen anomaly corresponds to the digits 2, 3, 4, 5, 6, 7, 8, and 9. For all datasets, we used class 0 as unseen anomaly, and select one normal class from the remaining 9 classes. The training dataset consists of 5,000 samples, of which 4,500 samples are unlabeled normal data points, 250 samples are labeled seen anomalies, and 250 samples are unlabeled seen anomalies. That is, the unlabeled data points in this dataset are contaminated with seen anomalies. We

used 10% of the training dataset as the validation dataset. The test dataset consists of 2,000 samples, about half of which are normal and the rest are anomalies, including both seen and unseen anomalies. More specifically, normal data points are sampled from the normal class in the test dataset, with a maximum of 1,000 samples, while both seen and unseen anomalies are sampled from their respective classes, with a maximum of 500 samples each. The example of the MNIST dataset is provided in Appendix A.

Next, we explain the last four datasets. CIFAR100 is just like CIFAR10 but consists of 100 classes. These classes are grouped into 20 superclasses, from which we used nature-related classes as normal and human-related classes as anomalies. We used the people class as unseen anomaly. PathMNIST, OCTMNIST, and TissueMNIST are medical image datasets. PathMNIST is colorectal cancer histology dataset with 9 tissue types. OCTMNIST is retinal optical coherence tomography dataset with 4 diagnostic categories. TissueMNIST kidney cortex cell dataset with 8 categories. For PathMNIST and TissueMNIST, we used the first class as unseen anomaly, selected one class from the remaining ones as normal, and treated the rest as seen anomaly. For OCTMNIST, since a pre-defined normal class exists, we used it as normal, used the first class as unseen anomaly, and treated the remaining classes as seen anomaly.

### 5.2. Methods

We compared our PUAE and PUSVDD with the following unsupervised and semi-supervised approaches.

**Unsupervised approaches:** We used the IF (Liu et al., 2008) as the shallow approach, and used the AE (Hinton & Salakhutdinov, 2006) and the DeepSVDD (Ruff et al., 2018) as the deep approaches. We also used the LOE (Qiu et al., 2022), which is robust to the anomalies in the unlabeled data. We chose the DeepSVDD as the base detector for the LOE.

**Semi-supervised approaches:** We used the ABC (Yamanaka et al., 2019), the DeepSAD (Ruff et al., 2019), and the SOEL (Li et al., 2023) that is a semi-supervised extension of the LOE. We chose the DeepSVDD as the base detector for the SOEL. We also used the PU learning binary classifier (PU) (Kiryo et al., 2017) for reference.

### 5.3. Setup

First, we outline the setups for all approaches except the IF. We used convolutional neural networks for the encoder and the decoder in the AE, the feature extractor in the DeepSVDD, and the binary classifier in the PU. The network architecture follows (Ruff et al., 2018). For the AE-based and DeepSVDD-based approaches, we set the dimension of the latent variable to 128. For the DeepSVDD-based

Table 1: Comparison of anomaly detection performance on the first four datasets.

| | MNIST | FashionMNIST | SVHN | CIFAR10 |
|---|---|---|---|---|
| IF (Liu et al., 2008) | 0.885 ± 0.062 | 0.916 ± 0.077 | 0.501 ± 0.014 | 0.610 ± 0.095 |
| AE (Hinton & Salakhutdinov, 2006) | 0.912 ± 0.042 | 0.841 ± 0.102 | 0.562 ± 0.040 | 0.535 ± 0.120 |
| DeepSVDD (Ruff et al., 2018) | 0.937 ± 0.045 | 0.921 ± 0.088 | 0.582 ± 0.035 | 0.709 ± 0.054 |
| LOE (Qiu et al., 2022) | 0.945 ± 0.033 | 0.916 ± 0.094 | 0.624 ± 0.054 | 0.718 ± 0.062 |
| ABC (Yamanaka et al., 2019) | 0.916 ± 0.042 | 0.841 ± 0.104 | 0.562 ± 0.041 | 0.535 ± 0.119 |
| DeepSAD (Ruff et al., 2019) | 0.942 ± 0.041 | 0.928 ± 0.089 | 0.652 ± 0.034 | 0.726 ± 0.051 |
| SOEL (Li et al., 2023) | 0.965 ± 0.026 | 0.936 ± 0.079 | 0.727 ± 0.043 | 0.775 ± 0.057 |
| PU (Kiryo et al., 2017) | 0.962 ± 0.034 | 0.922 ± 0.092 | 0.681 ± 0.096 | 0.693 ± 0.120 |
| PUAE | 0.983 ± 0.015 | 0.918 ± 0.085 | 0.689 ± 0.060 | 0.667 ± 0.066 |
| PUSVDD | **0.989 ± 0.012** | **0.948 ± 0.079** | **0.747 ± 0.080** | **0.803 ± 0.046** |

Table 2: Comparison of anomaly detection performance on the last four datasets.

| | CIFAR100 | PathMNIST | OCTMNIST | TissueMNIST |
|---|---|---|---|---|
| IF (Liu et al., 2008) | 0.604 ± 0.004 | **0.809 ± 0.118** | 0.714 ± 0.004 | 0.472 ± 0.187 |
| AE (Hinton & Salakhutdinov, 2006) | 0.589 ± 0.010 | 0.605 ± 0.240 | **0.860 ± 0.005** | 0.468 ± 0.178 |
| DeepSVDD (Ruff et al., 2018) | 0.587 ± 0.026 | 0.759 ± 0.148 | 0.726 ± 0.052 | 0.661 ± 0.055 |
| LOE (Qiu et al., 2022) | 0.576 ± 0.035 | 0.721 ± 0.160 | 0.783 ± 0.030 | 0.635 ± 0.086 |
| ABC (Yamanaka et al., 2019) | 0.590 ± 0.010 | 0.604 ± 0.241 | **0.857 ± 0.001** | 0.472 ± 0.177 |
| DeepSAD (Ruff et al., 2019) | 0.594 ± 0.012 | 0.763 ± 0.187 | **0.823 ± 0.038** | 0.683 ± 0.053 |
| SOEL (Li et al., 2023) | **0.633 ± 0.013** | 0.791 ± 0.145 | **0.856 ± 0.016** | 0.703 ± 0.062 |
| PU (Kiryo et al., 2017) | 0.541 ± 0.025 | **0.807 ± 0.132** | 0.614 ± 0.109 | 0.633 ± 0.082 |
| PUAE | 0.623 ± 0.014 | **0.776 ± 0.168** | **0.847 ± 0.011** | 0.594 ± 0.104 |
| PUSVDD | **0.637 ± 0.017** | **0.831 ± 0.152** | **0.857 ± 0.017** | **0.731 ± 0.077** |

approaches, we used no bias terms in each layer, pre-trained these feature extractors as the AE, and set the center **c** in Eq. (14) to the mean of the outputs of the encoder. We trained all methods by using Adam (Kingma & Ba, 2015) with a mini-batch size of 128. We set the learning rate to $10^{-4}$ and the maximum number of epochs to 200. We also used the weight decay (Goodfellow et al., 2016) with $10^{-3}$ and used early-stopping (Goodfellow et al., 2016) based on the validation dataset. We set the hyperparameter $\alpha = 0.1$ for the PUAE, the PUSVDD, the PU, the LOE, and the SOEL, which is the probability of anomaly occurrence.

Next, we outline the setup for the IF. We used the scikit-learn implementation (Pedregosa et al., 2011) and kept all hyperparameters at their default values in our experiments.

We trained unsupervised approaches using the unlabeled data[2], while we trained semi-supervised approaches using the unlabeled data and the labeled anomaly data.

---

[2]Note that we did NOT use the labeled anomaly data since unsupervised approaches cannot effectively use them.

To measure the detection performance, we calculated the AUROC scores for all datasets. We ran all experiments five times while changing the random seeds.

The machine specifications used in the experiments are as follows: the CPU is AMD EPYC 9124 16-Core Processor, the memory size is 512GB, and the GPU is NVIDIA RTX 6000 Ada.

### 5.4. Results

Tables 1 and 2 compare the anomaly detection performance on each dataset. We showed the average of the AUROC scores for all normal classes. We used bold to highlight the best results and statistically non-different results according to a pair-wise $t$-test. We used 5% as the p-value.

First, we focus on unsupervised approaches. The IF and the AE show significant performance variations across different datasets. Although the IF performed well on PathMNIST and the AE performed well on OCTMNIST, their performance became poor on other datasets. The DeepSVDD

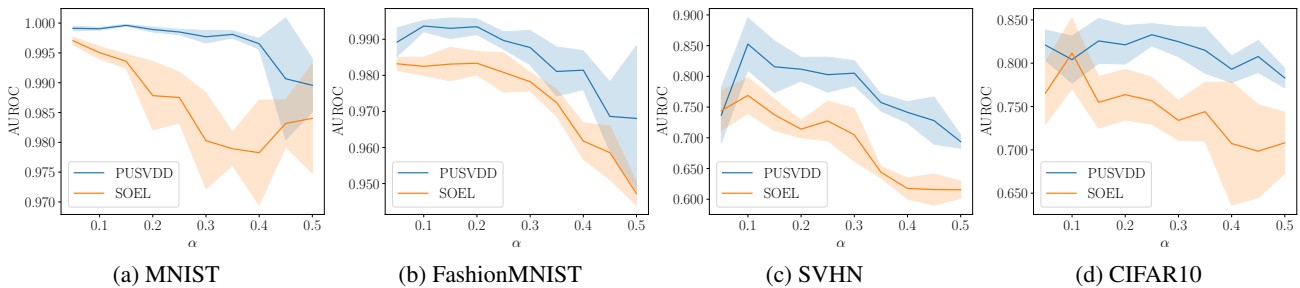

Figure 2: Relationship between the anomaly detection performance and the hyperparameter $\alpha$ of the PUSVDD and the SOEL on each dataset. We used class 0 as unseen anomaly, class 1 as normal, and the remaining classes as seen anomalies. The semi-transparent area represents standard deviations.

generally outperformed the IF and the AE, and the LOE, which is based on the DeepSVDD, often performed better than the DeepSVDD. However, since the LOE estimates anomalies within the unlabeled data in an unsupervised manner, incorrect estimations can lead to degraded performance.

Next, we focus on semi-supervised approaches. The ABC and the DeepSAD performed equal to or slightly better than the AE and the DeepSVDD, respectively. The reason for this is that ABC and DeepSAD assume that the unlabeled data are not contaminated with anomalies, which weakens the effect of maximizing anomaly scores for the labeled anomaly data. On the other hand, the SOEL outperformed DeepSAD in all cases. This is because the SOEL is capable of handling anomalies present in the unlabeled data.

Finally, we focus on the proposed methods. In most cases, the PUAE and the PUSVDD performed better than the AE and the ABC, the DeepSVDD and the DeepSAD, respectively. Especially, the PUSVDD achieved the best performance across all datasets. These results strongly indicate the effectiveness of our framework, which integrates PU learning with deep anomaly detection models.

In addition, we focus on the difference between the proposed methods and PU. The proposed methods performed equal to or better than the PU in all datasets. The reason is as follows. Since the PU is for binary classification, it sets the decision boundary between normal data points and seen anomalies. This prevents us from detecting unseen anomalies. On the other hand, our approach can detect unseen anomalies since it can model normal data points by the anomaly detector. The detection performance for seen and unseen anomalies are in Appendix C.

### 5.5. Hyperparameter Sensitivity

Our approach and the SOEL have the hyperparameter $\alpha$, which represents the probability of anomaly occurrence. In the above experiments, we set it to $\alpha = 0.1$ since the 10% of the training dataset is anomalies. Finally, we evaluate the

sensitivity of $\alpha$. Figure 2 shows the relationship between the anomaly detection performance and the hyperparameter $\alpha$ of the PUSVDD and the SOEL on each dataset.

In most datasets, the PUSVDD and the SOEL achieve the best performance around $\alpha = 0.1$. This indicates that, as with conventional PU learning, it is important to set $\alpha$ accurately. Note that $\alpha$ can be estimated from the unlabeled and anomaly training data using conventional PU learning approaches (Menon et al., 2015; Ramaswamy et al., 2016; Jain et al., 2016; Christoffel et al., 2016).

Compared to the SOEL, the PUSVDD is more robust to variations in $\alpha$. In other words, even if $\alpha$ deviates from the true value, the PUSVDD maintains relatively stable performance. This is because $\alpha$ in the SOEL is closely related to the number of anomalies within the unlabeled data. If $\alpha$ deviates from the true value, normal data may be incorrectly treated as anomalies, or vice versa.

On the other hand, since $\alpha$ in the PUSVDD only adjusts the weight of the loss function, it is expected to be relatively robust to deviations in $\alpha$.

## 6. Conclusion

Although most unlabeled data are assumed to be normal in semi-supervised anomaly detection, they are often contaminated with anomalies in practice, which prevents us from improving the detection performance. To solve this, we propose the deep positive-unlabeled anomaly detection framework, which integrates PU learning with deep anomaly detection models such as the AE and the DeepSVDD. Our approach enables us to approximate the anomaly scores for normal data with the unlabeled data and labeled anomaly data. Therefore, without the labeled normal data, we can train the anomaly detector to minimize the anomaly scores for normal data, and to maximize those for the anomaly data. Our approach achieves better detection performance than existing approaches on various image datasets. In the future, we will extend our approach to time-series data.

## Impact Statement

This paper presents work whose goal is to advance the field of Machine Learning. There are many potential societal consequences of our work, none which we feel must be specifically highlighted here.

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

## A. Dataset Example

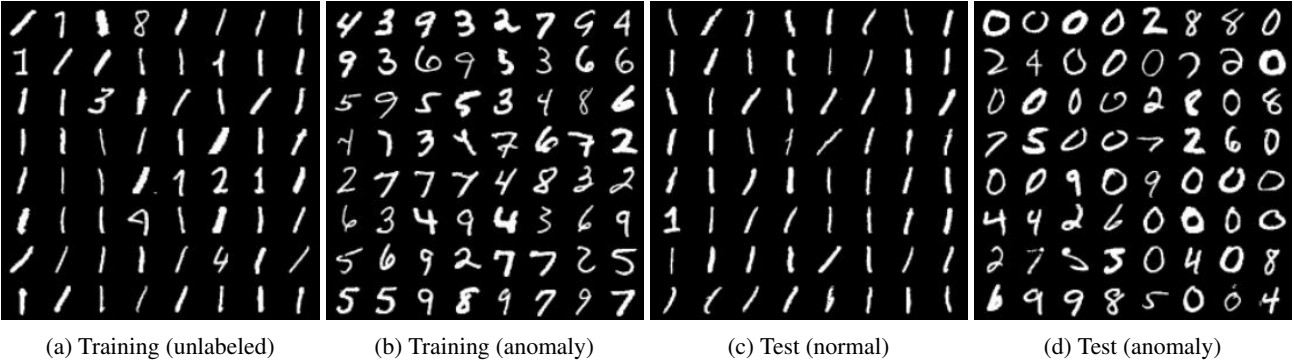

(a) Training (unlabeled)     (b) Training (anomaly)     (c) Test (normal)     (d) Test (anomaly)

Figure 3: The example of the dataset in the case of MNIST.

Figure 3 shows the example of the dataset in the case of MNIST. In this example, the digit 1 is normal, the digit 0 is unseen anomaly, and the digits 2, 3, 4, 5, 6, 7, 8, and 9 are seen anomalies. (a) The unlabeled data points in the training dataset are contaminated with seen anomalies. (b) The anomaly data points in the training dataset contain seen anomalies but not unseen anomalies. (c) The normal data points in the test dataset are not contaminated with anomalies. (d) The anomaly data points in the test dataset contain both seen and unseen anomalies.

## B. Anomaly Detection Performance with Various Numbers of Unlabeled Anomalies

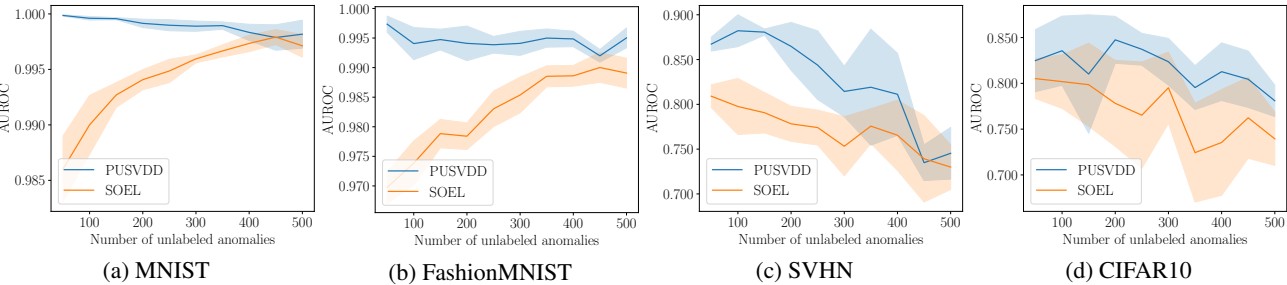

(a) MNIST     (b) FashionMNIST     (c) SVHN     (d) CIFAR10

Figure 4: Comparison of anomaly detection performance between the PUSVDD and the SOEL with various numbers of unlabeled anomalies on each dataset. We used class 0 as unseen anomaly, class 1 as normal, and the remaining classes as seen anomalies. The semi-transparent area represents standard deviations.

Figure 4 shows the anomaly detection performance with various numbers of unlabeled anomalies. The hyperparameter $\alpha$ was set to its true value for each case. Our PUSVDD achieved equal to or better performance than the SOEL.

## C. Anomaly Detection Performance for Seen and Unseen Anomalies

Tables 3, 4, 5 and 6 show the anomaly detection performance for seen and unseen anomalies, respectively. We showed the average of the AUROC scores for all normal classes. We used bold to highlight the best results and statistically non-different results according to a pair-wise $t$-test. We used 5% as the p-value. For seen anomalies, the PUSVDD achieved the best performance among all approaches. This shows the effectiveness of our approach, which is robust to the contaminated unlabeled data according to PU learning. For unseen anomalies, although the performance is highly dataset-dependent, the PUSVDD generally performs well. This indicates that we may be able to improve the detection performance for unseen anomalies by using seen anomalies. These results also show the difference between the conventional PU learning and our approach. The PU achieved the poor detection performance for unseen anomalies. This is because that it sets the decision boundary between normal data points and seen anomalies. On the other hand, our approach can detect unseen anomalies since it is based on the anomaly detector.

Table 3: Comparison of anomaly detection performance for seen anomalies on image datasets.

| | MNIST | FashionMNIST | SVHN | CIFAR10 |
|---|---|---|---|---|
| IF (Liu et al., 2008) | $0.814 \pm 0.096$ | $0.915 \pm 0.053$ | $0.510 \pm 0.016$ | $0.585 \pm 0.106$ |
| AE (Hinton & Salakhutdinov, 2006) | $0.843 \pm 0.073$ | $0.840 \pm 0.092$ | $0.563 \pm 0.039$ | $0.573 \pm 0.131$ |
| DeepSVDD (Ruff et al., 2018) | $0.925 \pm 0.056$ | $0.942 \pm 0.040$ | $0.600 \pm 0.033$ | $0.694 \pm 0.069$ |
| LOE (Qiu et al., 2022) | $0.942 \pm 0.038$ | $0.940 \pm 0.042$ | $0.645 \pm 0.055$ | $0.713 \pm 0.077$ |
| ABC (Yamanaka et al., 2019) | $0.852 \pm 0.072$ | $0.841 \pm 0.092$ | $0.564 \pm 0.039$ | $0.572 \pm 0.131$ |
| DeepSAD (Ruff et al., 2019) | $0.930 \pm 0.052$ | $0.956 \pm 0.032$ | $0.674 \pm 0.031$ | $0.716 \pm 0.074$ |
| SOEL (Li et al., 2023) | $0.967 \pm 0.024$ | $0.963 \pm 0.028$ | $0.751 \pm 0.035$ | $0.773 \pm 0.073$ |
| PU (Kiryo et al., 2017) | $0.959 \pm 0.036$ | $0.943 \pm 0.057$ | $0.678 \pm 0.099$ | $0.703 \pm 0.154$ |
| PUAE | $0.980 \pm 0.017$ | $0.942 \pm 0.041$ | $0.716 \pm 0.053$ | $0.695 \pm 0.083$ |
| PUSVDD | $\mathbf{0.994 \pm 0.004}$ | $\mathbf{0.972 \pm 0.029}$ | $\mathbf{0.787 \pm 0.069}$ | $\mathbf{0.796 \pm 0.059}$ |

Table 4: Comparison of anomaly detection performance for unseen anomalies on image datasets.

| | MNIST | FashionMNIST | SVHN | CIFAR10 |
|---|---|---|---|---|
| IF (Liu et al., 2008) | $0.955 \pm 0.031$ | $\mathbf{0.917 \pm 0.111}$ | $0.492 \pm 0.018$ | $0.635 \pm 0.095$ |
| AE (Hinton & Salakhutdinov, 2006) | $0.981 \pm 0.013$ | $0.842 \pm 0.130$ | $0.560 \pm 0.045$ | $0.497 \pm 0.111$ |
| DeepSVDD (Ruff et al., 2018) | $0.950 \pm 0.044$ | $0.900 \pm 0.143$ | $0.564 \pm 0.043$ | $0.724 \pm 0.087$ |
| LOE (Qiu et al., 2022) | $0.949 \pm 0.034$ | $0.892 \pm 0.152$ | $0.602 \pm 0.058$ | $0.724 \pm 0.099$ |
| ABC (Yamanaka et al., 2019) | $0.980 \pm 0.014$ | $0.840 \pm 0.133$ | $0.559 \pm 0.046$ | $0.497 \pm 0.109$ |
| DeepSAD (Ruff et al., 2019) | $0.954 \pm 0.040$ | $0.900 \pm 0.153$ | $0.631 \pm 0.044$ | $0.735 \pm 0.078$ |
| SOEL (Li et al., 2023) | $0.963 \pm 0.033$ | $0.908 \pm 0.135$ | $\mathbf{0.702 \pm 0.061}$ | $0.778 \pm 0.087$ |
| PU (Kiryo et al., 2017) | $0.965 \pm 0.038$ | $0.901 \pm 0.134$ | $\mathbf{0.684 \pm 0.102}$ | $0.683 \pm 0.133$ |
| PUAE | $\mathbf{0.985 \pm 0.017}$ | $0.894 \pm 0.140$ | $0.662 \pm 0.078$ | $0.639 \pm 0.090$ |
| PUSVDD | $\mathbf{0.983 \pm 0.024}$ | $\mathbf{0.924 \pm 0.134}$ | $\mathbf{0.708 \pm 0.103}$ | $\mathbf{0.811 \pm 0.101}$ |

Table 5: Comparison of anomaly detection performance for seen anomalies on real datasets.

| | CIFAR100 | PathMNIST | OCTMNIST | TissueMNIST |
|---|---|---|---|---|
| IF (Liu et al., 2008) | $0.577 \pm 0.005$ | $0.657 \pm 0.211$ | $0.679 \pm 0.005$ | $0.443 \pm 0.189$ |
| AE (Hinton & Salakhutdinov, 2006) | $0.514 \pm 0.013$ | $0.562 \pm 0.253$ | $\mathbf{0.808 \pm 0.005}$ | $0.443 \pm 0.188$ |
| DeepSVDD (Ruff et al., 2018) | $0.623 \pm 0.036$ | $0.769 \pm 0.139$ | $0.702 \pm 0.043$ | $0.692 \pm 0.047$ |
| LOE (Qiu et al., 2022) | $0.624 \pm 0.035$ | $0.746 \pm 0.167$ | $0.750 \pm 0.031$ | $0.657 \pm 0.084$ |
| ABC (Yamanaka et al., 2019) | $0.516 \pm 0.014$ | $0.564 \pm 0.252$ | $\mathbf{0.805 \pm 0.002}$ | $0.448 \pm 0.187$ |
| DeepSAD (Ruff et al., 2019) | $0.628 \pm 0.042$ | $0.772 \pm 0.165$ | $0.798 \pm 0.032$ | $0.715 \pm 0.040$ |
| SOEL (Li et al., 2023) | $\mathbf{0.696 \pm 0.020}$ | $0.806 \pm 0.142$ | $\mathbf{0.825 \pm 0.017}$ | $0.739 \pm 0.044$ |
| PU (Kiryo et al., 2017) | $0.630 \pm 0.023$ | $\mathbf{0.847 \pm 0.097}$ | $0.565 \pm 0.089$ | $0.628 \pm 0.080$ |
| PUAE | $0.576 \pm 0.025$ | $0.745 \pm 0.171$ | $0.800 \pm 0.011$ | $0.613 \pm 0.094$ |
| PUSVDD | $\mathbf{0.700 \pm 0.021}$ | $\mathbf{0.826 \pm 0.137}$ | $\mathbf{0.822 \pm 0.016}$ | $\mathbf{0.764 \pm 0.044}$ |

Table 6: Comparison of anomaly detection performance for unseen anomalies on real datasets.

| | CIFAR100 | PathMNIST | OCTMNIST | TissueMNIST |
|---|---|---|---|---|
| IF (Liu et al., 2008) | $0.632 \pm 0.005$ | $\mathbf{0.961 \pm 0.055}$ | $0.785 \pm 0.004$ | $0.500 \pm 0.187$ |
| AE (Hinton & Salakhutdinov, 2006) | $0.665 \pm 0.009$ | $0.647 \pm 0.239$ | $\mathbf{0.965 \pm 0.005}$ | $0.493 \pm 0.170$ |
| DeepSVDD (Ruff et al., 2018) | $0.552 \pm 0.033$ | $0.749 \pm 0.200$ | $0.774 \pm 0.072$ | $0.629 \pm 0.069$ |
| LOE (Qiu et al., 2022) | $0.528 \pm 0.047$ | $0.696 \pm 0.205$ | $0.851 \pm 0.030$ | $0.613 \pm 0.093$ |
| ABC (Yamanaka et al., 2019) | $0.665 \pm 0.009$ | $0.644 \pm 0.241$ | $\mathbf{0.961 \pm 0.002}$ | $0.496 \pm 0.168$ |
| DeepSAD (Ruff et al., 2019) | $0.561 \pm 0.024$ | $0.755 \pm 0.242$ | $0.874 \pm 0.049$ | $0.651 \pm 0.075$ |
| SOEL (Li et al., 2023) | $0.570 \pm 0.034$ | $0.776 \pm 0.190$ | $0.918 \pm 0.021$ | $0.666 \pm 0.093$ |
| PU (Kiryo et al., 2017) | $0.452 \pm 0.039$ | $0.767 \pm 0.215$ | $0.712 \pm 0.154$ | $0.638 \pm 0.098$ |
| PUAE | $\mathbf{0.671 \pm 0.011}$ | $0.806 \pm 0.204$ | $0.941 \pm 0.011$ | $0.574 \pm 0.116$ |
| PUSVDD | $0.573 \pm 0.027$ | $0.835 \pm 0.215$ | $0.925 \pm 0.022$ | $\mathbf{0.697 \pm 0.114}$ |

