# OpenReview forum: "Deep Positive-Unlabeled Anomaly Detection for Contaminated Unlabeled Data"
_ICML.cc/2025/Conference — Submitted to ICML 2025_

### Official Review · Reviewer_mGvk · 2025-03-07

**Overall Recommendation:** 2

**Summary:**

This paper proposes the deep positive unlabeled anomaly detection framework, to address the contaminated training samples problem for semi-supervised anomaly detection. Several anomaly detection datasets, including MNIST, CIFAR10, CIFAR100, etc., are utilized to evaluate the effectiveness of the proposed method. The results present that the proposed method outperform some alternatives.

## Updates after rebuttal
After carefully reviewing all the comments and responses, I have decided to maintain my original scores. The primary reason is the lack of a comprehensive literature review. As I pointed out in my previous comments, there have been several notable works in the field of industrial anomaly detection. For instance, SoftPatch has made significant contributions to noisy anomaly detection, which is not an unsupervised method as claimed by the authors. Given this, it is essential to compare the proposed method with more up-to-date techniques. However, the most recent method cited in this paper is SOEL, which was published in 2023.

**Claims And Evidence:**

N/A

**Essential References Not Discussed:**

N/A

**Experimental Designs Or Analyses:**

N/A

**Methods And Evaluation Criteria:**

The compared methods are not up-to-date, with the most advanced method SOEL published in 2023. In other anomaly detection fields like industrial anomaly detection, the author may find more advanced methods for comparisons.

**Other Comments Or Suggestions:**

N/A

**Other Strengths And Weaknesses:**

Weaknesses:
1. The setting and motivation do not appear to be novel in the context of anomaly detection. For example, the noisy-AD (SoftPatch) setting in industrial anomaly detection has been previously explored. Therefore, the authors should provide a detailed comparison and discussion with such related work.
2. The proposed method seems to be a combination of Positive-Unlabeled (PU) learning and existing anomaly detectors, such as Variational Autoencoders (VAE). As a result, it lacks significant novelty.
3. The writing of the paper requires substantial improvement. For instance, the explanations of unsupervised anomaly detection and semi-supervised anomaly detection are unclear and need to be more precise and comprehensive.
4. The experiments are insufficient. The paper only reports the performance using two classical anomaly detectors: VAE and SVDD. Moreover, the chosen baselines may not represent the state-of-the-art (SOTA) methods in this field. A more extensive evaluation with additional SOTA methods would strengthen the validation of the proposed approach.

**Questions For Authors:**

N/A

**Relation To Broader Scientific Literature:**

N/A

**Theoretical Claims:**

N/A

---

> ### Author Rebuttal · Authors · 2025-03-28
>
> Thank you for your feedback, which we shall address below.
>
> > Weakness 1: The setting and motivation do not appear to be novel in the context of anomaly detection. For example, the noisy-AD (SoftPatch) setting in industrial anomaly detection has been previously explored. Therefore, the authors should provide a detailed comparison and discussion with such related work.
>
> We would like to emphasize that the problem setting we address is a well-established one in the field of semi-supervised anomaly detection.
> Our contribution lies in proposing a novel framework that integrates PU learning with deep anomaly detectors under this setting.
>
> Thank you very much for pointing us to SoftPatch.
> Upon reviewing the paper, we found that SoftPatch [1] is an unsupervised anomaly detection method, and therefore not directly comparable to our semi-supervised approach.
> Nevertheless, we agree that it is a relevant and interesting work, and we will include it in the related work section of the revised paper to provide a more comprehensive discussion.
>
> [1] Jiang, Xi, et al. "Softpatch: Unsupervised anomaly detection with noisy data." *Advances in Neural Information Processing Systems* 35 (2022): 15433–15445.
>
>
> > Weakness 2: The proposed method seems to be a combination of Positive-Unlabeled (PU) learning and existing anomaly detectors, such as Variational Autoencoders (VAE). As a result, it lacks significant novelty.
>
> First, we would like to clarify a minor misunderstanding:
> our method is based on Autoencoders (AE), not Variational Autoencoders (VAE).
>
> More importantly, existing anomaly detectors such as AE or DeepSVDD cannot be directly combined with PU learning,
> as PU learning is formulated for binary classification, while anomaly detectors typically follow a one-class formulation.
>
> Our contribution lies in bridging this methodological gap.
> As detailed in Section 3, we design a tailored risk estimator and training objective to effectively integrate PU learning with deep anomaly detection.
> We believe this design represents a key aspect of our novelty.
>
>
> > Weakness 3: The writing of the paper requires substantial improvement. For instance, the explanations of unsupervised anomaly detection and semi-supervised anomaly detection are unclear and need to be more precise and comprehensive.
>
> We provided explanations of the problem setting in Section 2.1,
> unsupervised anomaly detection in Section 2.2,
> and semi-supervised anomaly detection in Section 2.3.
>
> We sincerely apologize if any parts were unclear.
> To help us improve the clarity of the revised version,
> could you kindly let us know which aspects were difficult to follow?
> We would greatly appreciate your feedback and will revise the manuscript accordingly.
>
>
> > Weakness 4: The experiments are insufficient. The paper only reports the performance using two classical anomaly detectors: VAE and SVDD. Moreover, the chosen baselines may not represent the state-of-the-art (SOTA) methods in this field. A more extensive evaluation with additional SOTA methods would strengthen the validation of the proposed approach.
> > Methods And Evaluation Criteria: The compared methods are not up-to-date, with the most advanced method SOEL published in 2023. In other anomaly detection fields like industrial anomaly detection, the author may find more advanced methods for comparisons.
>
> We would like to clarify that SOEL (Li et al., 2023) is currently one of the state-of-the-art methods for semi-supervised anomaly detection under label contamination.
> SOEL is based on DeepSVDD (Ruff et al., 2018),
> a representative unsupervised anomaly detector.
> Accordingly, we include DeepSVDD and its extensions as baselines,
> including DeepSAD (Ruff et al., 2019), LOE (Qiu et al., 2022), and SOEL itself.
>
> In addition, AE (Hinton \& Salakhutdinov, 2006) remains a widely used baseline in the literature,
> and we also compare it with its semi-supervised extension, ABC (Yamanaka et al., 2019).
> For completeness, we further include Isolation Forest (Liu et al., 2008) as a representative shallow detector,
> along with a standard PU learning-based binary classifier (Kiryo et al., 2017).
>
> We believe that our evaluation is both up-to-date and comprehensive, covering state-of-the-art methods and their fundamental models.
> Our framework is compatible with various base detectors as long as their loss functions are non-negative and differentiable,
> and has shown empirical improvements when applied to AE and DeepSVDD.
> If there are particular methods you believe should be included in our comparison, we would be happy to take them into consideration.

---

### Official Review · Reviewer_FDoK · 2025-03-11

**Overall Recommendation:** 3

**Summary:**

This paper presents a deep positive-unlabeled anomaly detection framework designed to address the issue of contaminated unlabeled data in anomaly detection. The framework integrates PU learning with deep anomaly detection models such as autoencoders and deep support vector data descriptions. It enables the approximation of anomaly scores for normal data using both unlabeled and labeled anomaly data, allowing the training of anomaly detectors without labeled normal data by minimizing anomaly scores for normal data and maximizing them for labeled anomalies.
The main contributions of the paper are:
1. The introduction of the deep PU anomaly detection framework, which effectively handles unlabeled data contaminated with anomalies.
2. Experimental results across various datasets demonstrating that the proposed approach achieves better detection performance compared to existing methods

## update after rebuttal
The method's applicability to various data types and its generalization to unseen anomalies are now clearer. However, its limitations in handling distribution shifts remain a concern, as this issue is not adequately addressed in the paper. Therefore , I maintain my score of weak accept and encourage further exploration of this challenge in future work

**Claims And Evidence:**

The claims made in the submission are supported by clear and convincing evidence. The authors conduct experiments across various datasets to demonstrate that the proposed deep positive-unlabeled  anomaly detection framework achieves better detection performance compared to existing approaches. The results show the framework's effectiveness in handling contaminated unlabeled data and its strong performance in detecting both seen and unseen anomalies.

**Essential References Not Discussed:**

No

**Experimental Designs Or Analyses:**

The experimental designs in the paper are generally reasonable but have room for improvement. The authors conduct experiments across multiple standard image datasets, use the AUROC as an evaluation metric, and compare with various baseline methods.

However, the paper lacks sufficient details in the experimental settings, such as data preprocessing steps, specific parameters of the model architecture, and hyperparameter settings during training. These details are crucial for other researchers to reproduce the experiments and validate the results. It is recommended to provide more detailed experimental settings in the paper to ensure the reproducibility of the results and facilitate further research.

**Methods And Evaluation Criteria:**

The proposed methods and evaluation criteria make sense for the problem or application at hand. The authors introduce a deep positive-unlabeled (PU) anomaly detection framework that integrates PU learning with deep anomaly detection models to handle contaminated unlabeled data. They demonstrate the effectiveness and superiority of the proposed approach by using the AUROC as a reasonable evaluation metric across various standard datasets

**Other Comments Or Suggestions:**

No more

**Other Strengths And Weaknesses:**

The paper is well-structured with clear logic. The methodology and experimental sections are described in detail, making it easy for readers to understand and reproduce the results.However the paper introduces a new framework, the theoretical analysis could be more in-depth.

**Questions For Authors:**

1. The paper's core idea is to use PU learning to handle unlabeled data contaminated with anomalies, but can this method adapt to the data characteristics and anomaly detection needs of different fields in practical applications?   For example, can the framework remain robust when there's a certain distribution shift in the overall data?
2. Has the paper fully considered the issue of unknown anomaly detection?   Ideologically, can it further expand the modeling and recognition strategies for unknown anomalies to enhance the model's generalization when facing new anomalies?

**Relation To Broader Scientific Literature:**

The proposed deep positive-unlabeled (PU) anomaly detection framework aligns with the broader field of semi-supervised learning, where a small amount of labeled data is used in conjunction with a large amount of unlabeled data to improve model performance. Existing semi-supervised approaches, such as those using autoencoders and deep support vector data descriptions, have laid the groundwork for integrating labeled anomaly data with unlabeled data. The PU framework extends these methods by specifically addressing the issue of contaminated unlabeled data, a common problem in real-world scenarios.

**Theoretical Claims:**

The theoretical claims in the paper are reasonable. The authors propose the deep positive-unlabeled anomaly detection framework based on the assumption of PU learning, where the unlabeled data distribution is a mixture of normal and anomaly data distributions. By expressing the normal data distribution as a combination of the unlabeled and anomaly data distributions, the authors derive a new training objective function that minimizes the anomaly scores for normal data and maximizes them for labeled anomaly data. This theoretical framework is logically consistent and aligns with the foundations of PU learning.

---

> ### Author Rebuttal · Authors · 2025-03-27
>
> Thank you for your feedback, which we shall address below.
>
> > Experimental Designs Or Analyses: However, the paper lacks sufficient details in the experimental settings, such as data preprocessing steps, specific parameters of the model architecture, and hyperparameter settings during training. These details are crucial for other researchers to reproduce the experiments and validate the results. It is recommended to provide more detailed experimental settings in the paper to ensure the reproducibility of the results and facilitate further research.
>
> Thank you for the suggestion.
> While the current version includes many experimental details such as network architecture (following Ruff et al., 2018), optimizer settings, training schedule, and dataset splits,
> we agree that providing further clarification would improve reproducibility.
> We will revise the paper to include more complete descriptions of experimental settings, possibly in tabular form to enhance readability.
>
>
> > Question 1: The paper's core idea is to use PU learning to handle unlabeled data contaminated with anomalies, but can this method adapt to the data characteristics and anomaly detection needs of different fields in practical applications? For example, can the framework remain robust when there's a certain distribution shift in the overall data?
>
> Similar to other semi-supervised anomaly detection approaches such as SOEL,
> our current approach is not designed to explicitly handle distribution shifts.
> We agree that distribution shift is a critical real-world challenge,
> and we plan to address it in future work.
> Fortunately, several approaches have been presented for adapting PU learning under distribution shift [1, 2],
> and we will explore incorporating such approaches into our semi-supervised anomaly detection framework.
>
> [1] Hammoudeh, Zayd, and Daniel Lowd. "Learning from positive and unlabeled data with arbitrary positive shift." NeurIPS 2020.
> [2] Kumagai, Atsutoshi, et al. "AUC Maximization under Positive Distribution Shift." NeurIPS 2024.
>
>
> > Question 2: Has the paper fully considered the issue of unknown anomaly detection? Ideologically, can it further expand the modeling and recognition strategies for unknown anomalies to enhance the model's generalization when facing new anomalies?
>
> Although it is important to provide theoretical guarantees for detecting unseen anomalies,
> our current approach offers only empirical and qualitative support for detecting unseen anomalies.
>
> What our approach guarantees is that,
> when labeled anomalies are available,
> the anomaly detector can be trained robustly even if similar anomalies are also present in the unlabeled data.
> As for detecting unseen anomalies,
> the performance mainly depends on the base anomaly detector.
> Anomaly detectors can treat data far from normal data as anomalies.
> Therefore,
> as long as unseen anomalies lie far from the normal data, they can be detected.
>
> Empirically, we have also observed in Appendix C that
> we may be able to improve the detection performance for unseen anomalies by using seen anomalies.
> Providing theoretical guarantees for this is an important direction for our future work.

---

> > ### Comment · Reviewer_FDoK · 2025-04-07
> >
> > Thank you for your responses.   The method's applicability to various data types and its generalization to unseen anomalies are now clearer.   However, its limitations in handling distribution shifts remain a concern, as this issue is not adequately addressed in the paper.  Therefore , I maintain my score of weak accept and encourage further exploration of this challenge in future work

---

> > > ### Author Response · Authors · 2025-04-07
> > >
> > > Thank you for your response.
> > > As you pointed out, distribution shift is an important challenge, and we would like to address it in our future work.

---

### Official Review · Reviewer_eovC · 2025-03-13

**Overall Recommendation:** 2

**Summary:**

The paper presents a novel semi-supervised anomaly detection method to improve the anomaly detection performance in handling contaminated unlabeled data. It integrates PU learning with deep anomaly detection models such as AE and DeepSVDD. The proposed approach outperforms existing anomaly detection methods.

**Claims And Evidence:**

Most claims of this paper are well-supported, particularly in handling contaminated unlabeled data, improving detection accuracy, and building on a solid theoretical foundation in PU learning.

**Essential References Not Discussed:**

The paper has discussed most relevant references.

**Experimental Designs Or Analyses:**

The experimental design is well-structured and follows standard best practices. However, 1) the authors do not analyze how performance degrades when only a few labeled anomalies are available. 2) The compared methods are relatively outdated. Both the semi-supervised and unsupervised approaches used for comparison are from earlier works.

**Methods And Evaluation Criteria:**

The methods and evaluation criteria make sense for the problem being addressed, but the compared methods are outdated.

**Other Comments Or Suggestions:**

See questions.

**Other Strengths And Weaknesses:**

Strengths:
1. The paper presents a novel semi-supervised anomaly detection in handling contaminated unlabeled data.
2.The writing structure of the paper is very clear.

Weaknesses:
1. The compared methods are relatively outdated. Both the semi-supervised and unsupervised approaches used for comparison are from earlier works. Many new models may outperform those based on AE and DeepSVDD, so it is necessary to include newer baselines.
2. The generalization of the method has not been validated on commonly used industrial anomaly detection datasets, such as MVTec and VisA datasets.
3. The paper only evaluates AUROC at the image level. However, pixel-level AUROC and other evaluation metrics, such as PR-AUC, have not been validated.

**Questions For Authors:**

1. The compared methods are relatively outdated. Both the semi-supervised and unsupervised approaches used for comparison are from earlier works. Many new models may outperform those based on AE and DeepSVDD, so it is necessary to include newer baselines.
2. The generalization of the method has not been validated on commonly used industrial anomaly detection datasets, such as MVTec and VisA datasets.
3. The paper only evaluates AUROC at the image level. However, pixel-level AUROC and other evaluation metrics, such as PR-AUC, have not been validated.
4. What happens if fewer labeled anomalies are available? In many real-world tasks, labeled anomalies are extremely scarce. The authors do not analyze how performance degrades when only a few labeled anomalies are available.

**Relation To Broader Scientific Literature:**

The paper presents a novel semi-supervised method to improve the anomaly detection performance in handling contaminated unlabeled data.

**Theoretical Claims:**

The paper provides a theoretical justification for the proposed method.

---

> ### Author Rebuttal · Authors · 2025-03-27
>
> Thank you for your feedback, which we shall address below.
>
> > Question 1: The compared methods are relatively outdated. Both the semi-supervised and unsupervised approaches used for comparison are from earlier works. Many new models may outperform those based on AE and DeepSVDD, so it is necessary to include newer baselines.
>
> We would like to clarify that our experimental setup includes SOEL (Li et al., 2023),
> which is currently one of the state-of-the-art methods for semi-supervised anomaly detection under label contamination.
> SOEL is based on DeepSVDD (Ruff et al., 2018),
> a representative unsupervised anomaly detector.
> Accordingly, we include DeepSVDD and its extensions as baselines,
> including DeepSAD (Ruff et al., 2019), LOE (Qiu et al., 2022), and SOEL itself.
>
> In addition, AE (Hinton \& Salakhutdinov, 2006) remains a widely used baseline in the literature,
> and we also compare it with its semi-supervised extension, ABC (Yamanaka et al., 2019).
> For completeness, we further include Isolation Forest (Liu et al., 2008) as a representative shallow detector,
> along with a standard PU learning-based binary classifier (Kiryo et al., 2017).
>
> We believe that our evaluation is both up-to-date and comprehensive, covering state-of-the-art methods and their fundamental models.
> Our framework is compatible with various base detectors as long as their loss functions are non-negative and differentiable,
> and has shown empirical improvements when applied to AE and DeepSVDD.
> If there are particular methods you believe should be included in our comparison, we would be happy to take them into consideration.
>
>
> > Question 2: The generalization of the method has not been validated on commonly used industrial anomaly detection datasets, such as MVTec and VisA datasets.
>
> As practical benchmarks,
> we chose PathMNIST, OCTMNIST, and TissueMNIST from the MedMNIST datasets (https://medmnist.com/),
> which was also used in the SOEL paper.
> Although the name "MNIST" may be misleading,
> we emphasize that these datasets are real medical datasets.
>
> Our intention was to provide a fair comparison under the same conditions as SOEL,
> and our approach consistently outperforms SOEL on these datasets.
> In future work, we plan to include experiments on MVTec and VisA.
>
>
> > Question 3: The paper only evaluates AUROC at the image level. However, pixel-level AUROC and other evaluation metrics, such as PR-AUC, have not been validated.
>
> Our approach is designed for image-level anomaly detection,
> where the goal is to determine whether an entire image contains anomalies,
> rather than localizing them at the pixel level.
> Therefore, pixel-level AUROC is not applicable to our method,
> as it does not produce pixel-wise anomaly scores.
>
> We followed the SOEL paper in using image-level AUROC,
> which we believe is the most appropriate metric for evaluating this setting.
> In future work, we will consider including additional evaluation metrics suitable for image-level detection,
> such as image-level PR-AUC.
>
>
> > Question 4: What happens if fewer labeled anomalies are available? In many real-world tasks, labeled anomalies are extremely scarce. The authors do not analyze how performance degrades when only a few labeled anomalies are available.
>
> In our experiments, we used 250 labeled anomalies.
> For example, even when this number is reduced to 50, the superiority of our proposed method remains.
> When evaluated on MNIST, our PUSVDD achieved an AUROC of 0.994, while SOEL obtained 0.963.

---

### Official Review · Reviewer_3Jde · 2025-03-15

**Overall Recommendation:** 1

**Summary:**

This paper tackles the task of semi-supervised anomaly detection when unlabeled training data is contaminated with anomalies. Specifically, this paper leverages positive-unlabeled learning to estimate anomaly scores for normal and anomaly data for anomaly detection. The quantitative results demonstrate the superiority of the proposed method over competing approaches.

**Claims And Evidence:**

Yes

**Essential References Not Discussed:**

No

**Experimental Designs Or Analyses:**

Yes

**Methods And Evaluation Criteria:**

Yes

**Other Comments Or Suggestions:**

No

**Other Strengths And Weaknesses:**

Strengths:
1) This paper presents detailed preliminaries, aiding readers in understanding the proposed method.

2) This paper provides a detailed discussion of related work.

3) The authors provide a detailed experimental setup.

4) I appreciate that the authors conduct an analysis on various contamination rate.

Weaknesses:

1) I appreciate that the authors provide a qualitative experimental comparison on a toy dataset in Figure 1. However, I believe that conducting experiments on real datasets may be necessary and be possible which could enhance the credibility of the evidence. Furthermore, why does PU learning fail to detect unseen anomalies, while integrating PU learning with deep anomaly detectors enables the detection of such anomalies?

2) As a key contribution, why does integrating positive-unlabeled learning with deep anomaly detectors help the detection model effectively estimate anomaly scores for test data and detect anomalies that are unseen during training? The authors are expected to provide a comprehensive explanation of the proposed method’s effectiveness from both empirical and theoretical perspectives.

3) In the problem setup, the authors mention that unlabeled anomalies are similar to labeled anomalies. However, in practice, unlabeled anomalies and labeled anomalies are likely to originate from different distributions. How does the proposed method perform in this scenario?

4) The assumption that hyperparameter α is known during training is not appropriate, as anomaly contamination rate for unlabeled data is difficult to obtain in practical scenarios.

5) The authors mention that the proposed can be applied to self-supervised methods, how does it perform?

6) Can the proposed method be applied to tabular data? How is the performance?

**Questions For Authors:**

See the weakness part.

**Relation To Broader Scientific Literature:**

To the reviewer, the main contribution of this paper is leveraging the existing positive-unlabeled learning for tackling the task of semi-supervised anomaly detection when training unlabeled data is contaminated with anomalies.

**Theoretical Claims:**

There seems to be no theoretical claim in this paper.

---

> ### Author Rebuttal · Authors · 2025-03-27
>
> Thank you for your feedback, which we shall address below.
>
> > Weakness 1: ... conducting experiments on real datasets ...
>
> We provide a qualitative evaluation using a toy dataset in Figure 1,
> and quantitative evaluations using real datasets such as SVHN, CIFAR10/100, and Path/OCT/Tissue-MNIST in Section 5.
> Although the name "MNIST" may be misleading, these are real medical datasets,
> and were also used in the SOEL paper,
> which represents the state-of-the-art in this field.
>
> To better clarify behavior on real data,
> we will include qualitative evaluations on these datasets in the revised paper.
>
>
> > Weakness 1: Furthermore, why does PU learning fail to detect unseen anomalies, while integrating PU learning with deep anomaly detectors enables the detection of such anomalies?
> > Weakness 2: As a key contribution, why does integrating positive-unlabeled learning with deep anomaly detectors help the detection model effectively estimate anomaly scores for test data and detect anomalies that are unseen during training? ...
>
> As shown in Figure 1a,
> since conventional PU learning is designed for binary classification,
> its decision boundary lies between the seen anomalies and normal data.
> As a result, even if unseen anomalies are far from the normal data,
> they cannot be detected.
>
> In contrast, our approach integrates PU learning with deep anomaly detectors, combining
> (1) PU learning’s ability to approximate the normal distribution using unlabeled and anomaly data, and
> (2) the anomaly detector’s ability to treat data far from this normal data distribution as anomalies.
> As a result, our approach can detect unseen anomalies as long as they deviate from normal data distribution.
> Tables 4 and 6 demonstrate that our approach outperforms conventional PU learning in detecting unseen anomalies.
>
>
> > Weakness 3: ... unlabeled anomalies and labeled anomalies are likely to originate from different distributions ...
>
> When labeled and unlabeled anomalies originate from different distributions,
> our approach treats all unlabeled data as normal, similarly to ABC and DeepSAD.
> We emphasize that even in such cases,
> if the unlabeled anomalies are few in number and lie far from the unlabeled normal data,
> they can be detected due to the nature of the base anomaly detector.
>
> Our target scenario assumes human-in-the-loop feedback,
> where labeling a few clear anomalies in unlabeled data is feasible.
> Our goal is to use such limited supervision to improve anomaly detection performance in this practical and cost-effective setting.
>
> To handle stronger distribution shifts of anomalies,
> we plan to incorporate PU methods designed for selection bias [1, 2].
>
> [1] Kato, Masahiro, Takeshi Teshima, and Junya Honda. "Learning from positive and unlabeled data with a selection bias." International Conference on Learning Representations. 2019.
> [2] Wang, Xutao, et al. "PUE: Biased positive-unlabeled learning enhancement by causal inference." Advances in Neural Information Processing Systems 36 (2023): 19783–19798.
>
>
> > Weakness 4: The assumption that hyperparameter $\alpha$ is known during training is not appropriate ...
>
> We assume that the labeled and unlabeled anomalies come from the same distribution.
> Under this assumption, as described in Section 3.1,
> $\alpha$ can be estimated using existing PU learning methods (e.g., Menon et al., 2015).
>
> In practice,
> $\alpha$ reflects the proportion of unlabeled data that resembles the labeled anomalies.
> Therefore, in the scenario described in Weakness 3,
> where labeled and unlabeled anomalies come from different distributions,
> the estimated value of $\alpha$ would be small.
> In such cases, our approach behaves similarly to existing semi-supervised approaches such as ABC and DeepSAD.
>
> Moreover, even if $\alpha$ is not estimated accurately,
> our approach is empirically more robust to its value than existing baselines, as demonstrated in Figure 2.
>
>
> > Weakness 5: The authors mention that the proposed can be applied to self-supervised methods, how does it perform?
>
> Our main contribution lies in using PU learning to approximate the loss for normal data using only unlabeled data and labeled anomalies.
> Hence, as long as the loss function for normal data is defined,
> our approach can be applied to any anomaly detection models,
> as described in Section 3.3.
> For example, it can be applied to self-supervised methods such as MHRot (Hendrycks et al., 2019), NTL (Qiu et al., 2021), and ICL (Shenkar \& Wolf, 2021),
> which are mentioned in the LOE paper (Qiu et al., 2022).
> We plan to evaluate the performance of our approach when applied to self-supervised anomaly detection and include the results in the revised paper.
>
>
> > Weakness 6: Can the proposed method be applied to tabular data? How is the performance?
>
> Yes.
> We have confirmed that our approach performs well on KDD99,
> a well-known tabular dataset for network anomaly detection.
> We will include additional tabular experiments in the revised version.

---

> > ### Comment · Reviewer_3Jde · 2025-04-07
> >
> > Dear authors,
> >
> > Thanks for your rebuttal. However, I still have major concerns about the novelty of this paper. As the authors claimed, the main contribution lies in utilizing PU learning to estimate the normal distribution, but the PU learning is an existing method. Additionally, the ability to detect unseen anomalies is derived from existing anomaly detection techniques. Furthermore, the authors stated that the proposed method performed well on self-supervised methods and tabular data, but this is neither reflected in the manuscript nor adequately addressed in the rebuttal. Even if it does, the performance still stems from existing methods. Based on these points, I am afraid that I would keep my previous rating.

---

> > > ### Author Response · Authors · 2025-04-07
> > >
> > > Thank you for your continued feedback and the opportunity to clarify our contributions.
> > > We would like to respectfully address your concerns regarding the novelty and scope of our contributions.
> > >
> > > > However, I still have major concerns about the novelty of this paper. As the authors claimed, the main contribution lies in utilizing PU learning to estimate the normal distribution, but the PU learning is an existing method. Additionally, the ability to detect unseen anomalies is derived from existing anomaly detection techniques.
> > >
> > > While it is true that PU learning is a well-established technique,
> > > the novelty of our work lies in bridging PU learning and deep anomaly detection in the context of semi-supervised anomaly detection under label contamination,
> > > a setting that has not been sufficiently explored.
> > > As described in Section 3 of the paper,
> > > we propose a new training objective that allows deep anomaly detectors (which typically rely on one-class learning) to benefit from PU learning (which is designed for binary classification).
> > > This integration is technically non-trivial,
> > > as it requires aligning the assumptions and training dynamics of PU learning and one-class anomaly detection.
> > >
> > > Regarding the ability to detect unseen anomalies,
> > > we fully agree that it stems from the underlying anomaly detector's design.
> > > However, we demonstrate empirically in Tables 4 and 6 (and Appendix C) that the integration with PU learning improves the generalization performance to unseen anomalies compared to using PU or the anomaly detector alone.
> > > Our contribution is not to invent an entirely new detection mechanism,
> > > but rather to develop a robust training framework that works effectively under realistic conditions of contamination and limited supervision.
> > >
> > > > Furthermore, the authors stated that the proposed method performed well on self-supervised methods and tabular data, but this is neither reflected in the manuscript nor adequately addressed in the rebuttal. Even if it does, the performance still stems from existing methods.
> > >
> > > We would like to respectfully clarify a misunderstanding.
> > > In the manuscript and rebuttal, we did not state that our method "performed well" on self-supervised methods.
> > > Rather, we stated that our framework can be applied to self-supervised anomaly detection methods, as long as the loss over normal data is defined.
> > > This was motivated by the SOEL (Li et al., 2023) paper,
> > > which also discusses compatibility with methods such as MHRot, NTL, and ICL.
> > > We plan to evaluate our method on these self-supervised baselines in future work and include such results in an extended version of the paper.
> > >
> > > Regarding tabular data, we did conduct a preliminary experiment using the KDD99 dataset and confirmed that our approach achieves strong performance.
> > > We will include these results in the revised version of the paper.
> > >
> > > Our evaluation strategy is designed to ensure a fair and consistent comparison with SOEL,
> > > which also builds on existing anomaly detectors by applying semi-supervised learning.
> > > In particular, SOEL is based on DeepSVDD, a widely adopted unsupervised method.
> > > Accordingly, we include DeepSVDD and its extensions (DeepSAD, LOE, and SOEL) as baselines.
> > >
> > > We also evaluate against AE and its semi-supervised variant ABC,
> > > as well as Isolation Forest and a standard PU learning method.
> > > We believe that this constitutes a comprehensive and up-to-date set of comparisons,
> > > including both state-of-the-art methods and foundational models.
> > >
> > > We hope this clarifies that while our work builds upon existing components,
> > > its novelty lies in the unified framework and practical relevance, not in isolated algorithmic innovations.
> > > We respectfully ask that these contributions be considered in the final assessment.

---

### Decision · Program_Chairs · 2025-05-01

**Decision:**

Reject

**Comment:**

The work receives one positive rating and three negative ratings, and the rebuttal does not result in any change in the ratings.

The major concerns include lack of novelty, incomplete (especially more recent) baselines, and comparison on popular industrial anomaly detection benchmarks.

The reviewers also find the problem setup "unlabeled anomalies are similar to labeled anomalies" questionable. A relevant yet probably more practical setting would be a supervised anomaly detection in open-set environments. A strong justification of the problem setup and its differences/advantages to methods in relevant research lines are missing.

While one reviewer leans towards acceptance, the overall consensus skewed toward rejection.